# Geographic Origin and Genetic Characteristics of Japanese Indigenous Chickens Inferred from Mitochondrial D-Loop Region and Microsatellite DNA Markers

**DOI:** 10.3390/ani10112074

**Published:** 2020-11-09

**Authors:** Ayano Hata, Atsushi Takenouchi, Keiji Kinoshita, Momomi Hirokawa, Takeshi Igawa, Mitsuo Nunome, Takayuki Suzuki, Masaoki Tsudzuki

**Affiliations:** 1Laboratory of Avian Bioscience, Department of Animal Sciences, Graduate School of Bioagricultural Sciences, Nagoya University, Nagoya, Aichi 464-8601, Japan; hata.ayano@e.mbox.nagoya-u.ac.jp; 2Laboratory of Animal Breeding and Genetics, Graduate School of Biosphere Science, Hiroshima University, Higashi-Hiroshima, Hiroshima 739-8528, Japan; takeatsu@hiroshima-u.ac.jp; 3Japanese Avian Bioresource Project Research Center, Hiroshima University, Higashi-Hiroshima, Hiroshima 739-8528, Japan; tigawa@hiroshima-u.ac.jp; 4Avian Bioscience Research Center, Graduate School of Bioagricultural Sciences, Nagoya University, Nagoya, Aichi 464-8601, Japan; kinokei123@hotmail.com; 5Laboratory of Animal Genetics, Department of Applied Molecular Biosciences, School of Bioagricultural Sciences, Nagoya University, Nagoya, Aichi 464-8601, Japan; t0m.h1r0kawa@icloud.com; 6Amphibian Research Center, Hiroshima University, Higashi-Hiroshima, Hiroshima 739-8526, Japan

**Keywords:** livestock animals, phylogeny, population genetics, evolution, conservation

## Abstract

**Simple Summary:**

Chickens have long lived with humans as companion animals for religious, ornamental, food production, or entertainment purposes. Chickens cannot fly or move long distances by themselves, and their introduction history is closely related to the history of human migration and trade. Indigenous chicken breeds with unique genetic characteristics have been established according to their uses and habitat in the world. Many ancient trade routes across the Eurasian continent played an important role in bringing various chicken breeds to each area, and Japan was one of the eastern ends of trade routes 1000–2000 years ago. In this study, molecular phylogenetic analyses suggested that Japanese indigenous chickens originated mainly from China, with some originating from Southeast Asia. Population genetic analyses revealed that most Japanese indigenous chicken breeds possess unique genetic characteristics. Furthermore, genetic assessments of Japanese indigenous chickens will provide new insights into the dispersal history and genomic evolution of domestic chickens. Moreover, investigation of indigenous chicken genetic characteristics contributes to revealing the genomic evolution of chickens and discovering candidate genetic resources for developing highly productive chicken breeds in their habitat environment.

**Abstract:**

Japanese indigenous chickens have a long breeding history, possibly beginning 2000 years ago. Genetic characterization of Japanese indigenous chickens has been performed using mitochondrial D-loop region and microsatellite DNA markers. Their phylogenetic relationships with chickens worldwide and genetic variation within breeds have not yet been examined. In this study, the genetic characteristics of 38 Japanese indigenous chicken breeds were assessed by phylogenetic analyses of mitochondrial D-loop sequences compared with those of indigenous chicken breeds overseas. To evaluate the genetic relationships among Japanese indigenous chicken breeds, a STRUCTURE analysis was conducted using 27 microsatellite DNA markers. D-loop sequences of Japanese indigenous chickens were classified into five major haplogroups, A–E, among 15 haplogroups found in chickens worldwide. The haplogroup composition suggested that Japanese indigenous chickens originated mainly from China, with some originating from Southeast Asia. The STRUCTURE analyses revealed that Japanese indigenous chickens are genetically differentiated from chickens overseas; Japanese indigenous chicken breeds possess distinctive genetic characteristics, and Jidori breeds, which have been reared in various regions of Japan for a long time, are genetically close to each other. These results provide new insights into the history of chickens around Asia in addition to novel genetic data for the conservation of Japanese indigenous chickens.

## 1. Introduction

Chickens are some of the most widely used livestock animals worldwide. Since the beginning of domestication several thousand years ago, chickens have lived with humans as a companion animal for religious, ornamental, or entertainment purposes. Biologists have been interested in the origins of chicken domestication [1,2,3,4]. The main wild ancestor of chickens is the red junglefowl, *Gallus gallus*, which lives in tropical forests in Asia. In general, chickens are thought to have been domesticated in Southeast Asia earlier than 6000 BC [5]. Even though chicken domestication in Northern China during the early Holocene (~10,000 B.P.) was inferred from a 326-bp fragment of a mitochondrial D-loop region obtained from ancient “chicken” bones [6], recent studies have proposed a later beginning of chicken domestication based on morphological data for ancient bones of chicken-like species [7] and molecular phylogenetic analysis [8]. Mitochondrial D-loops also indicated that *G. g. gallus*, *G. g. spadiceus*, and *G. g. jabouillei* contributed to the domestication of modern chicken populations [9,10].

A large amount of research has examined the matrilineal history of domestic chickens using mitochondrial DNA (mtDNA) sequence data with a special focus on the control region (D-loop) because of its higher mutation rate than that in other mitochondrial regions. The mitochondrial D-loop sequences of chickens worldwide have been classified into five major haplogroups (A‒E), four minor haplogroups (F‒I), and six rare haplogroups (J, K, W‒Z) [10,11]. Haplogroups A and B prevail in Asian regions, including China, Japan, and South and Southeast Asia [10,11,12,13]. Haplogroups C and D are exhibited in populations from Africa, Madagascar, South to East Asia, and the Pacific islands [11,12,13,14,15]. Haplogroup E is broadly distributed in the Middle East, North Africa, Europe, and South America [11,12,14,16,17]. Haplogroups F and G are found to be restricted to Southwestern China and Southeast Asia [11,12,18]. Haplogroup H consists of chickens in restricted regions in southwestern China, Japan, and Thailand [11,19,20], and haplogroups I and J have been reported only for a few indigenous chickens in Northeast India and Southeast Asia [11,18,20] and wild red junglefowls in Southeast Asia [11,20]. Haplogroups K and W‒Z have been rarely found in wild red junglefowls in Northeast India and Southwestern China, respectively [11].

Japanese chickens are designated as chicken breeds that were established and reared by the Meiji era (1868–1912). Nearly 50 breeds are known as indigenous breeds [21], and 17 breeds (two breed groups and 15 breeds) are assigned as natural monuments of Japan. These breeds have been bibliographically classified into three groups [22]: Jidori, Shokoku, and Shamo. Jidori chickens, which are believed to be the oldest among the three groups, were introduced into Japan more than 2000 years ago and have been reared as various local breeds across Japan. Shokoku is thought to have been introduced into Japan from China about 1000 years ago. Shokoku was initially used for cock-fighting for a while after its introduction and was then used as a base breed to produce several ornamental chicken breeds because of its beautiful plumage, including long tail feathers and saddle hackles. Shamo seems to have been brought into Japan about 1000–500 years ago for the purpose of chicken fighting [23]. This chicken breed has an upright posture, long neck, and long and thick legs, similar to those of chicken breeds used for cock-fighting throughout Southeast Asia. Various chicken breeds have been established since the 17th century by crossing the three chicken groups described above, or importing chicken breeds from abroad, such as Ukokkei (based on Silkie) and Chabo (possibly based on chickens from Vietnam). Most of the existing Japanese indigenous chickens were established during the Meiji Era (1868–1912) [21]. To understand the genetic relationships of Japanese indigenous chickens, biochemical markers, such as blood group systems and blood proteins, were examined for chicken breeds in Japan at the end of the 20th century [24,25,26,27,28,29]. Two routes for the introduction of Japanese indigenous chickens have been assumed based on previous morphological and blood group system studies [30,31]; one is a northern route from China via the Korean peninsula, and the other is a southern route from Southeast Asia through islands extending from Northern Taiwan to Okinawa and Southwestern Kyushu. The genetic relationships of Japanese indigenous chickens have also been examined using DNA markers. Komiyama et al. (2003, 2004) [23,32] revealed three clades, A, B, and C, in the mitochondrial D-loop region of fighting cocks (Shamo) and long-crowing chicken breeds. Oka et al. (2007) [33] subdivided the D-loop sequences of 20 Japanese indigenous chicken breeds into seven groups. However, the phylogenetic relationships of D-loop sequences between Japanese indigenous chickens and chickens worldwide require further clarification. The genetic characteristics of Japanese indigenous chickens have also been examined using autosomal microsatellite DNA markers [34,35,36]. Osman et al. (2006) [34] investigated the genetic relationships of 22 Japanese indigenous chicken breeds, including 17 natural monument breeds, and found that their genetic relationships were more complicated than the traditional framework described in Oana (1951) [22]. Tadano et al. (2007, 2008) [35,36] examined the genetic relationships of nine long-tail Japanese indigenous chicken breeds and seven miniature Japanese indigenous chicken breeds, in which approximately 40 individuals were collected from multiple populations of each breed. These studies revealed genetic differences in Japanese indigenous chickens at the breed level but did not shed light on genetic variations at individual or population levels. Since some Japanese indigenous chickens consist of various populations that are reared by different farmers, fanciers, and livestock centers, genetic differentiation might have accumulated among such populations. Thus, another issue that remains is the extent of the genetic differences among individuals or populations within breeds. Additionally, since chickens are one of the most important livestock for humans for multiple purposes, are not migratory birds, and have a small home range [37], their dispersal history is thought to be closely related to the histories of human trade and movement [14,38,39,40]. Historically, Japan is located on the eastern end of the ancient trade network across the Eurasian continent, and various chicken breeds have been introduced from throughout Asia. Thus, phylogenetic analyses and genetic characterization of Japanese indigenous chickens will provide new insights into the dispersal history and genomic evolution of chickens worldwide. In this study, we aimed to reveal: (1) the genetic relationships between Japanese indigenous chickens and chickens worldwide, and (2) the genetic relationships between breeds, focusing on genetic variation within breeds using mitochondrial D-loop sequences and 27 microsatellite DNA markers.

## 2. Materials and Methods

### 2.1. Ethics Statement

Animal care and all experimental procedures were conducted according to the guidelines for the care and use of experimental animals of Nagoya University. The animal protocols were approved by the Animal Experiment Committee of the Graduate School of Bioagricultural Sciences, Nagoya University.

### 2.2. Sample Collection and Genomic DNA Extraction

In this study, 38 Japanese indigenous chicken breeds were investigated. The breakdown of the breeds is as follows: 24 Japanese natural monument breeds (Chabo (CHB), Echigo Nankin Shamo (ENK), Gifu-Jidori (GJI), Hinai-dori (HNI), Iwate-Jidori (IJI), Jitokko (JTK), Kawachi-Yakko (KWC), Koeyoshi (KYS), Ko-Shamo (KSM), Kurokashiwa (KKS), Mie-Jidori (MJI), Minohiki-dori (MNH), Ohiki (OHK), Oh-Shamo (OSM), Onaga-dori (ONG), Satsuma-dori (STM), Shokoku (SHK), Tomaru (TMR), Tosa-Jidori (TSJ), Totenko (TTK), Ukokkei (UKK), Uzurao (UZR), Yamato Gunkei (YMG), and Yakido (YKD)), 10 breeds which are not recognized as natural monuments but are well known as Japanese indigenous chickens (Aidu-Jidori (AJI), Chahn (CHN), Kinpa (KNP), Kumamoto (KMM), Kureko-dori (KRK), Nagoya (NGY), Ryujin-Jidori (RJI), Sadohige-Jidori (SJI), Tokudi-Jidori (TKJ), and Tosa Kukin (TSK)), and four breeds which are recognized as Japanese indigenous chickens in the Japan Agricultural Standards (Cochin (CCN), Ingie (ING), Barred Plymouth Rock (BPR), and Rhode Island Red (RIR)).

In addition, we investigated seven indigenous chicken breeds from overseas (Araucana (ARC), Black Minorca (BMN), Brahma (BRM), Lite Sussex (LSX), Belgian Mille Fleur bantam (MIL), White Leghorn-LA (WLL), and White Leghorn-MK (WLM)), and a population of wild ancestor, red junglefowl (RJF). The origins of these breeds are as follows: ARC: South America; BMN, LSX, MIL, WLL and WLM: Europe; BRM: South Asia; and RJF: Southeast Asia. In total, 955 Japanese native chicken individuals were examined in this study, consisting of 38 Japanese native chicken breeds and seven chicken breeds overseas (Table 1). All 955 blood samples that were used for genomic DNA extraction were provided by the Graduate School of Biosphere Science at Hiroshima University. Genomic DNA was extracted from 20 μL whole blood using commercial DNA extraction kits, ISOSPIN Blood and Plasma DNA (NIPPON GENE, Tokyo, Japan). These samples were provided from Hiroshima University, Nagoya University, chicken fanciers, and livestock experimental stations in each area/prefecture (Table 1).

### 2.3. Sequencing of the mtDNA D-Loop Region and Genotyping of Microsatellite Markers

Partial DNA fragments (500-bp) of the mtDNA D-loop region were amplified by PCR for all 38 Japanese indigenous chicken breeds. The D-loop region of four chicken breeds overseas, Araucana, Black Minorca, Brahma, and Lite Sussex, were also examined as an outgroup of the Japanese breeds. The following primer sets were used: Gg_Dloop_1F (5′-AGGACTACGGCTTGAAAAGC-3′) [41] and Gg_Dloop_4R (5′-CGCAACGCAGGTGTAGTC-3′) [33]. Amplification was performed in a 10-μL reaction mixture containing 50 ng genomic DNA, 10 pmol of each primer, and 5.0 μL of SapphireAmp^R^ Fast PCR Master Mix (TaKaRa, Shiga, Japan). The cycling conditions were as follows: initial denaturation at 94 °C for 1 min, followed by 35 cycles at 94 °C for 20 s, 58 °C for 5 s, and 72 °C for 10 s, and a final extension for 5 min at 72 °C. The PCR products were detected by electrophoresis on 1.5% agarose gels and then purified using a 20% polyethylene glycol/2.5-M NaCl precipitation method [42,43]. The cycle sequencing reaction was performed using a Big Dye^TM^ Terminator Cycle Sequencing Kit v3.1 (Applied Biosystems, Foster City, CA, USA), and nucleotide sequences were determined using an ABI PRISM 3130 genetic analyzer (Applied Biosystems).

Twenty-seven microsatellite markers, which were selected from the 30 markers recommended for chicken biodiversity studies by the Food and Agriculture Organization [44] (Appendix A) were used in this study. PCR amplification was performed using a 10-μL reaction mixture containing approximately 50 ng genomic DNA, 2 μL of dNTP mix, 5 μL of 2 × PCR buffer for KOD FX Neo, 3 pmol of each primer, and 0.5 units of KOD FX Neo DNA polymerase. The cycling conditions were as follows: initial denaturation at 94 °C for 2 min, followed by 35 cycles at 98 °C for 10 s, 60 °C for 30 s, and 68 °C for 30 s, and a final extension for 5 min at 68 °C. PCR products were electrophoresed with Hi-Di formamide (Applied Biosystems) and GeneScan 600 LIZ Size Standard (Applied Biosystems) using the ABI PRISM 3130 genetic analyzer (Applied Biosystems). Allele sizes were determined using GENEIOUS PRIME v2019.2.1 (Biomatters Ltd., Auckland, New Zealand).

### 2.4. Phylogenetic Analysis of mtDNA D-Loop Sequences

DNA sequences were aligned using the Geneious Alignment method implemented in GENEIOUS PRIME v2019.2.1 (Biomatters Ltd.). Ambiguous sites behind the primer sequences were trimmed from the fragments. To determine the phylogenetic positions of Japanese indigenous chickens in chicken populations worldwide, Bayesian phylogenetic trees were constructed using BEAST v2.4.3 [45], including 420 D-loop sequences (A01, A02, …Y, and Z) obtained from GenBank, which were defined by Miao et al. (2013) [11] (Appendix A). The best-fit substitution model of the D-loop sequences was determined based on the Bayesian Information Criterion using jModeltest v2.1.10 [46,47]. The Bayesian phylogenetic analysis was performed using 10 million Markov Chain Monte Carlo (MCMC) generations, sampling a tree every 1000 generations. The convergence of the runs was verified using Tracer v1.7.1 [48]. After discarding the first 10% of the sampled 10,000 trees as burn-in, a maximum clade credibility tree was constructed from the remaining trees using Tree Annotator v2.4.3 [49]. FigTree v1.4.2 was used to illustrate the summarized tree. Haplogroup I was used as an outgroup according to the phylogenetic tree of Miao et al. (2013) [11].

### 2.5. Estimation of Haplogroup Frequencies Worldwide

D-loop sequences of *G. gallus* with nucleotide lengths ranging from 400 bp to 1300 bp were retrieved from the NCBI GenBank via the Ebot system. Among the 8348 sequences retrieved, 6529 sequences of domestic chickens, which were registered with information for their “country” data or breed names, were selected and their geographic origins were determined (Appendix A). Haplogroup assignments for the 6529 sequences were performed using the MitoToolPy program [50], together with 420 D-loop sequences (A01, A02, …Y, and Z, Appendix A) used in the Bayesian phylogenetic analysis. Then, haplogroup frequencies in nine areas of the world (Europe; Africa; West, Central, South, Southeast, and East Asia; and North and South America) were examined.

### 2.6. Analysis of Genetic Diversity and Genetic Structures Based on mtDNA D-Loop Sequences

The nucleotide diversity (*pi*) [51], number of haplotypes (*H*), and Watterson estimator per sequence (*Theta-w*) [52] were calculated using DnaSP v5 [53]. The demographic history of each population was estimated using Tajima’s *D* test [54] implemented in DnaSP v5.

### 2.7. Polymorphisms in Microsatellite DNA Markers

Genetic diversity indices, which are allelic richness (AR), mean number of alleles per population (*Na*), null allele frequency (*NAF*), and *F* statistics (*F_IS_*, *F_ST_*, and *F_IT_*), were calculated for each microsatellite DNA marker using MICROSATELLITE ANALYSER v4.05 [55] (*AR*, *F_IS_*, *F_ST_*, and *F_IT_*), GenAlEx v6.5 [56] (*Na*), or Cervus v3.0.7 [57,58] (*NAF*) (Appendix A).

### 2.8. Genetic Diversity of 46 Chicken Breeds Based on Microsatellite Markers

Subsequently, the genetic diversity indices were examined for each of the 37 Japanese indigenous chicken breeds and eight indigenous chicken breeds from foreign countries. The YMG breed, which consisted of two individuals, was excluded from the population genetic analyses because not enough microsatellite markers could be amplified (less than 20 markers). MICROSATELLITE ANALYSER 4.05 was used to calculate *AR*, GenAlEx 6.5 was used to calculate *MNA* and the mean number of effective alleles (*Ne*), and Arlequin v3.5.2.2 [59] was used to determine the observed (*Ho*) and expected heterozygosity (*He*) (Appendix A). Null allele frequencies for each locus and breed were estimated using FreeNA [60]. Deviation from Hardy–Weinberg equilibrium at each locus and in each breed was tested using Arlequin v3.5.2.2. [59]. Pairwise *Fst* genetic distances between breeds were calculated using MICROSATELLITE ANALYSER v4.05 [55].

Bayesian clustering analysis was performed using STRUCTURE v2.3 [61] to infer genetic clustering patterns for all 45 chicken breeds and for the 37 Japanese indigenous chicken breeds. The log probability values from K = 1 to K = 46 for the 45 breeds and from K = 1 to K = 38 for the 37 Japanese indigenous chicken breeds were estimated using the admixture model and the correlated allele frequency model [62]. The length of the MCMC generations for data sampling was 100,000, which started after a burn-in period of 100,000 generations. Twenty independent MCMC runs were operated for each K. Of the 20 MCMC runs of each K, any MCMC runs with variances of log likelihood more than twice those of the other MCMC runs were excluded from subsequent analyses. The clustering patterns of the remaining runs were analyzed to generate one major clustering pattern in each K using clumpak [63]. Subsequently, the optimal K value was determined using the Evanno method [64] in Structure Harvester v0.6.94 [65].

## 3. Results

### 3.1. D-Loop Haplotypes of Japanese Indigenous Chicken Breeds

We determined the nucleotide sequences of 500-bp fragments of the mtDNA D-loop region, including the hypervariable segment I, for 669 Japanese native chickens from 38 breeds, and in 57 individuals of four chicken breeds from foreign countries. A total of 37 haplotypes were identified, 36 of which were possessed by the Japanese indigenous chickens, and Hap_37 was found in five individuals of LSX (Table 2, accession nos. LC575218–LC575942, Appendix A). The 36 haplotypes of the Japanese indigenous chickens were temporally classified into five common haplogroups, A, B, C, D, and E (Figure 1), according to the haplotype groups defined previously by Liu et al. (2006) [10] and Miao et al. (2013) [11]. Among the Japanese indigenous chicken breeds, haplogroups A and E were the main groups, followed by C, D, and B. Five and 14 haplotypes were classified into haplogroups A and E, consisting of 231 individuals from 26 breeds and 165 individuals from 25 breeds, respectively. Nine haplotypes found in 147 individuals from 20 breeds belonged to haplogroup C. Seven haplotypes of haplogroup D were discovered in 75 individuals from 14 breeds. Only 51 individuals from eight breeds had Hap_11 in haplogroup B. Of the 37 haplotypes, 27 haplotypes were identical to the haplotypes determined in Miao et al. (2013) [11], and the remaining 10 haplotypes (Hap_22‒24, Hap_26, Hap_28, Hap_30, Hap_31, and Hap_35‒37) showed one or two base differences from the haplotypes, although they were rarely observed. Haplotypes A01 and A05 were the most frequently observed haplotypes, and were found in 100 individuals from 16 breeds and 107 individuals from 12 breeds, respectively. Haplotype E01 was the third most frequently observed haplotype, and was found in 91 individuals from 15 breeds. Haplotype C01 was the most frequently observed haplotype of haplogroup C (11 breeds, 65 individuals), followed by C06 (seven breeds, 38 individuals) and C07 (nine breeds, 27 individuals). Among the seven haplotypes of haplogroup D, the haplotypes that were exhibited by more than two breeds were D04 (3 breeds, 19 individuals), D06 (five breeds, 17 individuals), and D13 (eight breeds, 30 individuals).

### 3.2. Genetic Diversity of mtDNA D-Loop Sequences of Japanese Indigenous Chickens

ENK and TSK were examined for only one individual and therefore were excluded from the genetic diversity analyses. Among the Japanese indigenous chicken breeds, the number of D-loop haplotypes in each population ranged from 1 (AJI, ENK, IJI, KMM, KNP, RJI, TSK and ING) to 11 (CHB and OSM) (Table 3), showing the ranges of haplotype diversity (Theta-w) from 0.000 to 6.000 (2.992 on average) and nucleotide diversity (pi) from 0.000 to 0.014 (0.007 on average). The eight breeds (CHB, JTK, KKS, MJI, OSM, SHK, STM, and UUK) exhibited relatively higher genetic diversity (*pi*, 0.012 for CHB, JTK, SHK, STM, and UKK to 0.014 for KKS; *Theta-w*, 4.070 for STM to 5.693 for MJI). Regarding the indigenous chicken breeds overseas, ARC and LSX exhibited similar levels of genetic diversity to Japanese indigenous chickens (*Theta-w*, 3.145 for LSX and 3.335 for ARC; *pi*, 0.005 for LSX and 0.006 for ARC). BRM and BMN had one and two haplotypes, respectively. Ten out of the 38 Japanese indigenous chicken breeds (CHN, HIN, MNH, NGY, OHK, ONG, SJI, TKJ, UZR, and CCN) exhibited negative Tajima’s D values, and of those, NGY and ONG were significant (*p* < 0.05), suggesting that the chickens were bred under purifying selection within each population.

### 3.3. Distribution of D-Loop Haplogroups Worldwide

The 7527 sequences from 43 geographic locations (Appendix A) were classified into nine haplogroups (A, B, C, D, E, F, G, H/I/K, and W-Z), and the frequencies of the nine haplogroups in nine global regions were assessed (Figure 2): Japan (*n* = 938), East Asia (China and South Korea, *n* = 3392), Southeast Asia (*n* = 791), South Asia (*n* = 764), West and Central Asia (*n* = 171), Europe (*n* = 835), Africa (*n* = 245), North America (*n* = 15), and South America (*n* = 370). East Asia displayed all nine haplogroups, and haplogroups A, B, and E were the three major haplogroups there. Southeast Asia harbored eight haplogroups, but haplogroups B and D were the first and second most frequently observed haplogroups. South Asia and regions west of that area showed haplogroup E as their predominant haplogroup.

### 3.4. Genetic Characteristics of Indigenous Chicken Breeds Estimated Using 27 Microsatellite DNA Markers

Six-hundred and eighty-nine Japanese indigenous chickens of 37 breeds, 122 chickens from seven indigenous chicken breeds of other countries, and 20 individuals of red junglefowls were utilized for genetic diversity analyses using microsatellite markers. The allelic richness (*AR*) values ranged from 1.347 for MCW0098 to 1.913 for LEI0192 (1.699 on average) (Appendix A). *Na* ranged from 0.870 for MCW0014 to 4.093 for LEI0192 (2.637 on average). *F_IS_* varied from –0.127 for MCW0103 to 0.448 for MCW0014 (0.080 on average). The *F_ST_* and *F_IT_* values fell within the range of 0.358 (MCW0103) to 0.869 (MCW0014), and 0.276 (MCW0103) to 0.928 (MCW0014), respectively (*F_ST_* = 0.476, *F_IT_* = 0.511 on average). The null allele frequencies were higher than 0.2 at 25 of the 27 loci (Appendix A). Looking at population level, the number of loci that showed null allele frequencies higher than 0.2 ranged from 1 to 10 (1.7 on average). Significant departures from Hardy-Weinberg equilibrium were observed in one population for each of the four markers: MCW0081, MCW0183, MCW0330, and MCW0165 (*p* < 0.05) (Appendix A).

Out of the 34 Japanese indigenous chicken breeds examined, except for four populations composed of less than one individual (ENK, TSK, YKD, and YMG), RJI exhibited the least genetic diversity (*AR* = 1.263; mean number of alleles (*Na*) = 1.593; *Ne* = 1.386; *Ho* = 0.184) (Table 3). In the other Japanese indigenous chicken breeds, *AR* ranged from 1.284 (MJI) to 1.646 (UKK), *Na* ranged from 1.815 (KNP) to 6.222 (OSM), *Ne* ranged from 1.522 (KNP) to 3.081 (UKK), and *Ho* ranged from 0.254 (MJI) to 0.565 (KMM). In the indigenous chicken breeds overseas (except for RJF), *AR* ranged from 1.260 (BRM) to 1.417 (WLL); *Na* ranged from 1.630 (MIL) to 2.778 (WLL), *Ne* ranged from 1.465 (LSX) to 1.931 (WLL), and *Ho* ranged from0.148 (BMN) to 0.402 (WLL).

### 3.5. Genetic Relationships among Japanese Indigenous Chicken Breeds and Indigenous Chicken Breeds from Other Countries

The cladogram based on the *Fst* genetic distances constructed with 27 microsatellite markers is shown in Figure 3. Five Jidori breeds, GJI, IJI, TKJ, RJI, and KRK, were divided into the same group, whereas four other Jidori were phylogenetically located far from these five breeds. Two Jidori breeds, MJI and TSJ, showed close genetic relationships. Two other Jidori breeds, SJI and AJI, each constructed independent clades with CHN and KMM, respectively. OSM and KSM had a close relationship. Three Shamo-related breeds, KNP, YKD, and ENK, were separated from OSM and KSM, but clustered in the same clade. While SHK, MNH, and KKS showed close genetic relationships, other Shokoku-related (OHK, ONG, and TMR) species were phylogenetically separated from each other. Two meat production breeds (HNI and NGY) fell in the same cluster as CCN, which is also used for meat production.

Genetic clustering patterns of Japanese indigenous chickens determined by STRUCTURE Analysis using STRUCTURE HARVESTER (http://taylor0.biology.ucla.edu/structureHarvester/) revealed that K = 2 was optimal for the 45 breeds (delta K = 17.23), in which Japanese indigenous chicken breeds and indigenous chicken breeds from overseas were clearly separated (Figure 4a,b). Although four Japanese indigenous chicken breeds (ING, BPR, CCN, and RIR) were classified into the same cluster as the indigenous chicken breeds from overseas, these four breeds have a long history of being kept in Japan and are recognized as Japanese indigenous chicken breeds. In addition, the STRUCTURE analysis of the 38 Japanese indigenous chicken breeds indicated K = 2 as the optimal number of clusters (delta K = 5.42) (Figure 4c); however, we focused on the clustering patterns after K = 3 because the clustering patterns at K = 2 were similar to those for the 45 breeds described above (Figure 4d). The 37 Japanese indigenous chicken breeds were classified into three clusters at K = 3; seven breeds of Jidori chicken (AJI, GJI, IJI, MJI, RJI, TKJ, and TSJ) and KRK were classified into the same group (navy), and the Shamo and Shokoku-related groups were clustered into the same group (light blue). Other breeds were shown in orange or constructed mixed clusters. At K = 5, it was shown that the breeds that are closely related to Shamo or Shokoku shared the characteristics of the population (dark green and purple). The Jidori chicken breeds from K = 3 to K = 16 were classified into the same group, except for SJI, and KRK was also grouped with the six Jidori chicken breeds. SJI appeared in the same group as CHN from K = 3 to K = 21. At K = 3 and K = 5, three dual-purpose breeds (for egg and meat production) (NGY, KMM, and RIR) consisted of the same cluster as other indigenous chicken breeds from overseas, and from K = 16 to K = 37, NGY shared the same color as RIR (red) and KMM was clustered with CCN (purple).

## 4. Discussion

### 4.1. Genetic Relationships of Japanese Indigenous Chickens with Chickens Worldwide

To date, the mitochondrial D-loops of 23 Japanese indigenous chicken breeds have been examined [9,11,23,32,33,66,67] (Miyake, 2000 Direct submission to GenBank; Sakahira, 2007 Direct submission to GenBank; Wada et al., 2008 Direct submission to GenBank), and 24 breeds have been examined using microsatellite DNA markers [34,35,36]. Genetic diversities among the Japanese indigenous chicken breeds were well researched in the previous researches, while genetic relationships of the indigenous chicken breeds to chicken breeds overseas remained unveiled. In this study, we performed genetic characterization of 38 Japanese indigenous chicken breeds consisting of 669 individuals using mitochondrial D-loop region and 689 individuals from 37 breeds for 27 microsatellite DNA markers. As indicated by Oka et al. (2007) [33], our mitochondrial D-loop sequences also suggested that Japanese indigenous chicken breeds were not clearly subdivided into three groups, Jidori, Shokoku, and Shamo, which were proposed by Oana et al. (1951) [22]. This was the first study to reveal the phylogenetic relationships between Japanese indigenous chickens and chickens worldwide. To understand the geographic origins of D-loop sequences of Japanese indigenous chickens, we reconstructed the global haplogroup distribution as surveyed in a previous study [11] because the haplogroup data worldwide have been expanded in the last few years. Among the 4732 D-loop sequences of domestic chickens examined in the previous study, 2732 sequences (57.7%) were from East Asia, 88.7% of which (2426/2732) were from China. For Southeast Asia, which is a central area of the origin of chicken domestications, only 268 indigenous chickens (5.6%) were included. In this study, the number of sequences from Southeast Asia markedly increased to 761, accounting for 11.7% of our dataset. The global haplogroup distribution was almost the same as that in the previous study, but our study revealed that haplogroup D was distributed at higher frequencies in Southeast Asia than those shown by Miao et al. (2013) [11]. The haplogroups A, C, and E accounted for a high proportion of Japanese indigenous chickens. The haplogroup composition of the Japanese indigenous chickens appeared similar to the composition of chickens in East Asia. Conversely, even though haplogroup D was less frequently observed in East Asia (mainly China), a moderate frequency of haplogroup D was exhibited in Japan. Haplogroup D is distributed in Southeast Asia, Southeast Asian Islands, and Polynesia, and is thought to be the representative genotype of the “Pacific chicken” [14,40]. 

These results suggest a large impact of chicken introduction from China on the establishment of Japanese indigenous chickens; chickens from Southeast Asia also made a small, but important, contribution to this establishment. For example, the CHN chicken used in this study is an indigenous chicken on the Japanese southern island of Okinawa and had haplotype D04 in high frequency (16 of 21 individuals). The genetic contribution of Southeast Asian chickens to Japanese indigenous chickens in the southern part of Japan was also suggested by haplogroup H, which was discovered in Shamo chickens in the Japanese southern island of Okinawa by a previous study [23]. Komiyama et al. (2003) [23] hypothesized that the origin of Haplogroup H (described as “Group A” in their study) was China because Haplogroup H was not found in other areas in the world when the paper was published. Since their study, Haplogroup H has been found in Southern China, Vietnam, and Thailand [10,11,19,20] (Dich et al., 2011, Direct submission to GenBank). These results were congruent with previous hypotheses regarding the northern and southern introduction routes of Japanese indigenous chickens [30,31]. Another remarkable point found in our study was that haplogroup B, which was one of the largest haplogroups in East and Southeast Asia, was found less frequently in Japan, and only haplotype B01 was found in several chicken breeds. This result allows us to hypothesize that the introduction of haplogroup B into Japan occurred either once or a few times from a restricted chicken population in East or Southeast Asia. Several chicken breeds possessed haplotypes that differed by one or a few bases from the representative haplotypes of Miao et al. (2013) [11], such as A05_2 of ONG, as well as C01_2 and C01_3 of SHK. These two breeds are well-known for their long beautiful tail, and ONG is a unique breed whose tail feathers continue to grow without falling out throughout their life. This breed cannot be found in other countries, and ONG has been recognized as a special natural monument in Japan since 1952. Through continuous artificial selection to retain or strengthen their morphological characteristics, it is suggested that they have breed-specific haplotypes. To further understand the history of Japanese indigenous chickens, mitochondrial genome sequencing for representative haplotypes of haplogroups A–E and H is needed.

### 4.2. Genetic Diversity among Japanese Indigenous Chickens

OSM, CHB, and UKK showed higher genetic diversities than those of the other Japanese indigenous chicken breeds, even though we were unable to examine large number of individuals for several indigenous chicken breeds. In addition to their large sample size, the reason for their high genetic diversity could be their breeding background. Oh-shamo is the traditional breed used for cock-fighting game in Japan, and this breed was suggested to have been introduced from China and Southeast Asia [23]. In addition, Oh-shamo has been frequently crossed among many populations to make more aggressive and stronger chickens. Chabo is a breed comprising many varieties of plumage colors, leading to high genetic diversity within the breed. Ukokkei, whose English name is Silkie, had relatively higher genetic diversity among the Japanese indigenous chickens in a previous study by Osman et al. (2006) [34] using microsatellite DNA markers. Ukokkei is one of the 17 breeds of the National Monument of Japan, but Silkie is reared in various areas around Asia. Notably, numerous D-loop haplotypes (*h* = 9) were observed in UKK, and their frequencies were nearly equal to each other (from 1 to 3, except for 6 for haplotype C37; Table 2). These results suggest that Ukokkei was introduced from multiple source populations of Silkie in Asia, leading to high genetic diversity within breeds. Conversely, Jidori breeds, such as AJI and GJI, showed low genetic diversities. In particular, the remarkably low genetic diversity of RJI among the Jidori breeds was obvious, despite the number of individuals examined. This result was congruent with that of a previous study, in which many individuals from multiple populations were examined using 20 microsatellite markers [68]. Because our DNA samples of RJI were collected in 2012, which was before Oka et al. (2014) [68] proposed the necessity of conservation of Ryujin-jidori, further genetic monitoring for Ryujin-jidori as well as the other Jidori breeds is needed.

### 4.3. Genetic Relationships among Japanese Indigenous Chickens

The genetic relationships inferred from the phylogenetic tree of *F_ST_* genetic distance were partially similar but mostly different from those of Osman et al. (2006) [34], possibly due to the greater number of breeds and individuals examined in this study. In accordance with Osman et al. (2006) [34], our results also showed genetic similarity between CHB and UZR as well as JTK and STM. One of the new findings in our study was the close genetic relationships of most of the Jidori breeds. According to the STRUCTURE analysis, six breeds of Jidori (AJI, GJI, IJI, MJI, RJI, and TKJ) and KRK were classified into the same group from K = 3 to K = 16. This was an interesting finding in that Jidori breeds could maintain the common genetic characteristics of their ancestral breed, even though these breeds have been reared in different areas throughout Japan. Although close genetic relationships among the three Shamo-related breeds, KNP, KSM, and OSM, were evident in the *F_ST_* phylogenetic tree, TTK, which is a Shokoku-related breed showing long-crowing, was also included in the Shamo clade. The genetic similarity between OSM and TTK was exhibited in the STRUCTURE plots at K = 5 (purple cluster). Three Shokoku-related breeds, KKS, MNH, and SHK, formed a clade in the *F_ST_* phylogenetic tree, which supported the breeding histories of Kurokashiwa and Minohiki. Kurokashiwa is thought to be derived from Shokoku [22], and Minohiki is thought to be established by crossing Shokoku and Shamo [21]. Close genetic relationships were suggested for SJI and CHN by both the *F_ST_* tree and STRUCTURE plots from K = 3 to K = 21, even though they are reared in geographically distant places; SJI is in the Niigata Prefecture in Northern Japan, and CHN is in the Okinawa Prefecture. A possible explanation for the close relationships between SJI and CHN was that both breeds have beards, although we could not determine whether or not their “beard” originated from the same ancestral breed. From K = 5 and K = 16, three dual purpose breeds, NGY, KMM, and RIR, shared the same color. All three breeds were created by crossing Cochin chicken, and NGY and KMM have similar body shapes to Cochin. This result suggested that one of their origin breeds reflected the current genetic characteristics. In the STRUCTURE plot at K = 37, the Japanese indigenous chicken breeds mostly formed exclusively independent clusters. This result supports a previous study by Osman et al. (2006) [34], who suggested that each breed has been kept independently, avoiding genetic contamination among different breeds. Meanwhile, some indigenous chicken breeds, such as JTK, TSJ, and STM, exhibited genetic differences within populations, implying that genetic diversity can be recovered by cross-breeding among populations when such breeds suffer from inbreeding deficiencies.

## 5. Conclusions

Japanese indigenous chickens have a long history of hundreds, even thousands, of years in some breeds [21,22]. Even though genetic characteristics obtained in this study were only parts of the gene pool of Japanese indigenous chicken breeds, we could provide additional insights into the dispersal history of chickens around Asia in addition to novel genetic data for the conservation of Japanese indigenous chickens. Our study suggests that Japanese indigenous chickens were largely introduced via land routes through China and partially established from chickens introduced via sea routes from Southeast Asia. Close genetic relationships among the Jidori breeds were revealed in this study, an interesting result given that these breeds have been separately maintained in various parts of Japan for nearly 2000 years. Chickens cannot fly or move long distances by themselves, and their introduction history must be closely related to the history of human migration and trade. Therefore, understanding the genetic characteristics of Japanese indigenous chicken breeds will not only contribute to their genetic conservation, but also provide new or supportive knowledge of human history around Asia. Mitochondrial genome sequences of chickens, which have recently accumulated in GenBank, provide a new time framework for the introduction histories of chickens throughout the world. In addition, as previous studies have demonstrated [35,36,68,69], genetic assessment with microsatellite markers is an effective way to survey genetic diversity not only at the population level but also at the individual level, and to develop a valid strategy for recovering genetic diversity of an endangered breed.

## Figures and Tables

**Figure 1 animals-10-02074-f001:**
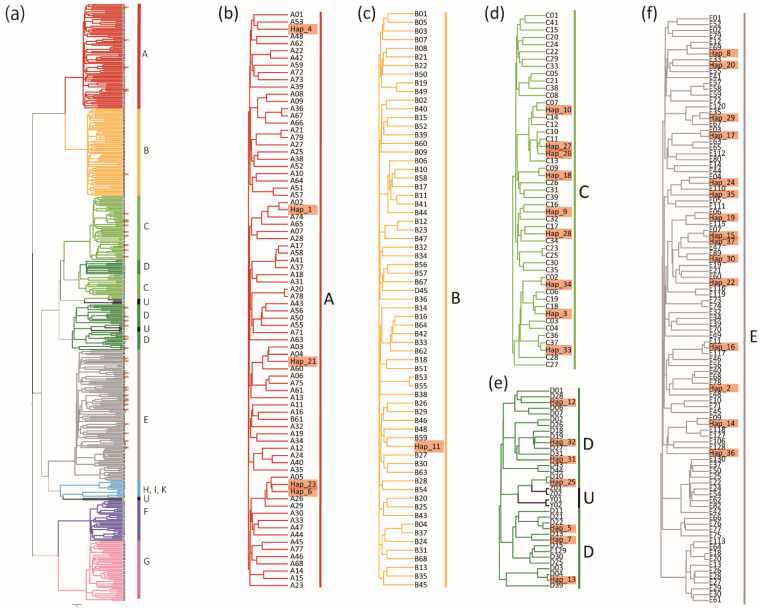
Bayesian phylogenetic tree of mitochondrial DNA D-loop haplotypes (Hap_1–Hap_37, highlighted with orange boxes) found in Japanese native chickens from 38 breeds, and in 57 individuals of four chicken breeds from foreign countries. The tree was constructed with 420 D-loop sequences (A01, A02, …Y, and Z), which were defined by Miao et al. (2013) (Appendix A). Five clades which included the 37 haplotypes are highlighted with light blue boxes in the whole phylogenetic tree (**a**). Each of the five clades is shown in a separate figure (**b**–**f**).

**Figure 2 animals-10-02074-f002:**
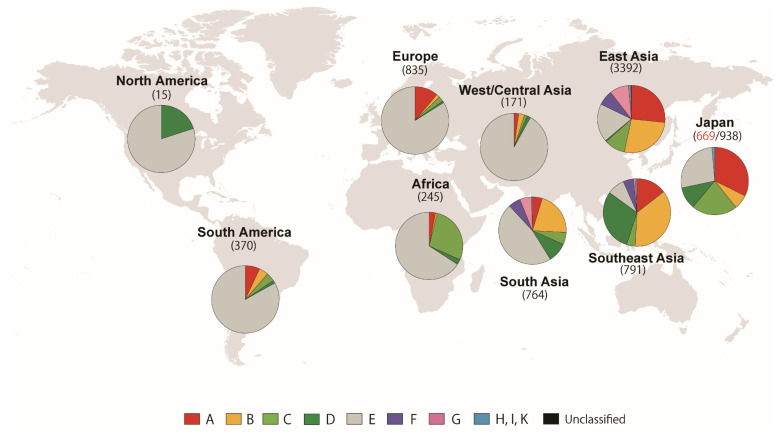
Distribution of D-loop haplogroups worldwide inferred from 6795 D-loop sequences obtained from GenBank and 669 sequences of Japanese indigenous chickens obtained in this study. The frequencies of nine haplogroups (A, B, C, D, E, F, G, H/I/K, and unclassified) in nine regions of the world are shown by pie charts. The number of sequences in each region is indicated in parentheses.

**Figure 3 animals-10-02074-f003:**
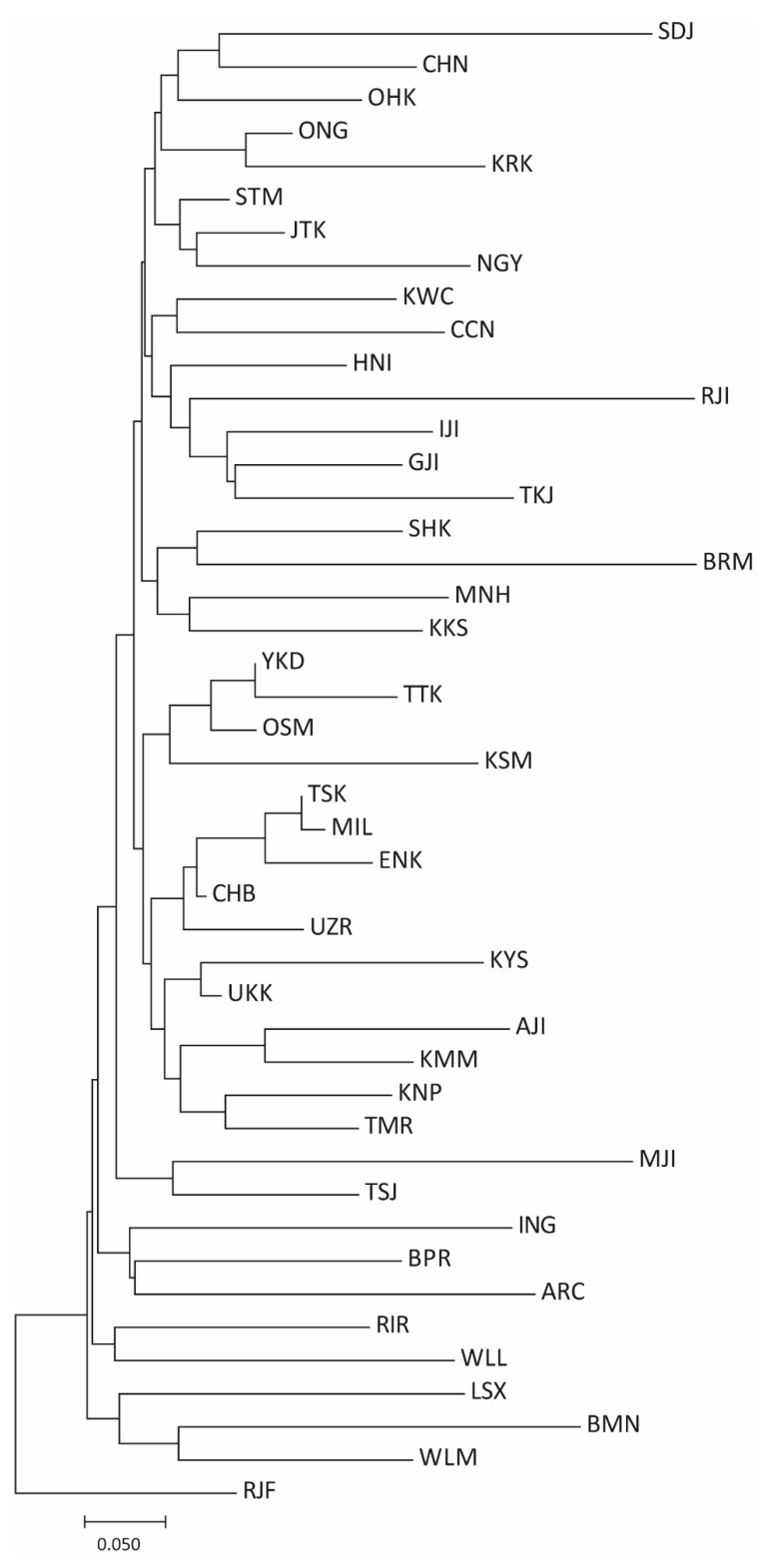
Phylogenetic relationships among 45 chicken breeds based on pairwise *F_ST_* genetic distance calculated with 27 microsatellite DNA markers.

**Figure 4 animals-10-02074-f004:**
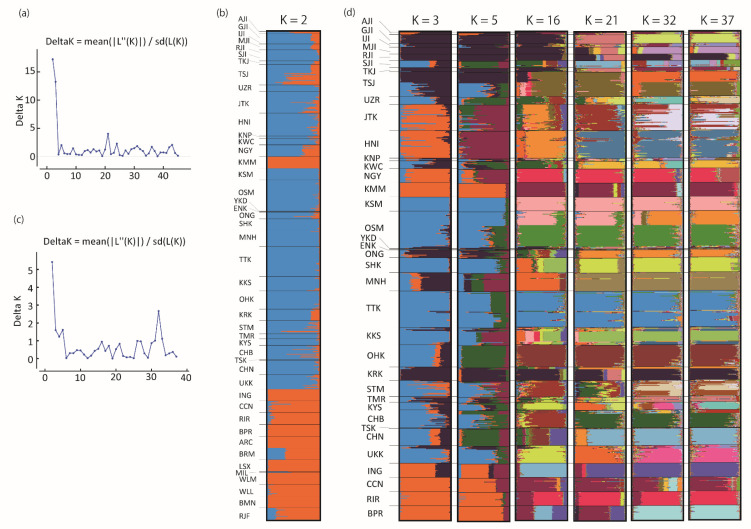
STRUCTURE plots of 37 Japanese indigenous chicken breeds, 7 chicken breeds from foreign countries, and one red junglefowl population. (**a**) Delta K values from K = 2 to K = 46 for all of 45 chicken breeds and populations. (**c**) Delta K values from K = 2 to K = 38 for 37 Japanese indigenous chicken breeds. (**b**) Group memberships to K = 2 clusters of 45 chicken breeds. (**d**) Group memberships to “K” clusters of 37 Japanese indigenous chicken breeds (K = 3, 5, 16, 21, 32, and 37). Genetic clusters are represented in different colors.

**Table 1 animals-10-02074-t001:** List of breeds of Japanese indigenous chickens and indigenous chickens overseas examined in this study.

Category	Breeds	Breed ID	No. of Individuals	Source of Samples †	D-Loop	Microsatellite
Japanese indigenous chicken	Aidu-Jidori	AJI	3	FPLES	3	3
	Chabo	CHB	64	Fukushima1-3, Kanagawa1-3, Mie1, Hiroshima1-3, Yamaguchi, IPLES, AZES, Shimane1-2	64	25
	Chahn	CHN	24	Okinawa, Osaka	21	24
	EchigoNankin	ENK	1	Hiroshima University	1	1
	Gifu-Jidori	GJI	13	IPLES, GPLRI, Hiroshima University, Fukuoka1	9	13
	Hinai	HNI	39	Fukushima4, APLES, TUA, Yamaguchi, AZES, Hiroshima University	30	39
	Iwate-Jidori	IJI	5	IPAHES	4	5
	Jitokko	JTK	37	Hiroshima1-2, Hiroshima University, Osaka, Miyazaki, Kagoshima, Yamaguchi	15	37
	Kawachi-Yakko	KWC	16	Osaka, Mie2-3, TUA, Hiroshima2, IPLES	16	11
	Kinpa	KNP	4	Fukushima4	4	4
	Koeyoshi	KYS	16	Osaka, Aomori, TUA, Yamaguchi	15	11
	Ko-Shamo	KSM	20	Hiroshima university	19	20
	Kumamoto	KMM	20	KLES	13	20
	Kureko-dori	KRK	23	Kumamoto	22	19
	Kurokashiwa	KKS	25	Hyogo1, Hiroshima1, Mie3, YPLES, Shimane1-5	14	25
	Mie-Jidori	MJI	10	Mie2-4, Hiroshima2	6	10
	Minohiki-dori	MNH	29	IPLES, Shizuoka1	29	26
	Nagoya	NGY	22	Hiroshima Univerisity	20	20
	Ohiki	OHK	31	Kochi, Shimane1-2, Kochi1-2	26	31
	Oh-Shamo	OSM	59	Fukushima1, KPLES, Hiroshima University, Kagoshima2, Ehime	59	53
	Onaga-dori	ONG	39	Kochi3-9	29	11
	Ryujin-Jidori	RJI	9	Wakayama1-4, Nara1	9	9
	Sadohige-Jidori	SJI	10	Hiroshima University	7	10
	Satsuma-dori	STM	23	Osaka, Nara2, TUA, Yamaguchi, Hiroshima2	18	23
	Shokoku	SHK	33	Osaka, Hyogo2, Hiroshima University, Mie2-5, Hiroshima2, TUA, Yamaguchi	25	21
	Tokudi-Jidori	TKJ	6	Yamaguchi	6	6
	Tomaru	TMR	14	Shizuoka2, TUA, NAHES, IPLES	14	8
	Tosa-Jidori	TSJ	35	Osaka, Fukushima4, Kochi1-2, Kochi10-11, KPLES	16	35
	Tosa-Kukin	TSK	1	Osaka	1	1
	Totenko	TTK	51	Kochi12, Osaka, Fukushima5, Mie3, Mie5, Hiroshima1-2, Ymaguchi, Kochi10	36	51
	Ukokkei	UKK	29	Aichi1, Fukushima4, Hiroshima1-2, Mie3, IPLES, AZES, MLRD, APLRC	29	25
	Uzurao	UZR	16	Kochi1-2, Kochi13-14	16	11
	Yakido	YKD	3	Mie4, Fukushima4	3	1
	Yamato Gunkei	YMG	2	Yamaguchi	2	0
	Cochin	CCN	20	Fukushima6	18	20
	Ingie	ING	20	Fukuoka2, Hiroshima University, Yamaguchi	19	20
	Rhoad Island Red	RIR	20	Aichi2	16	20
	Barred Plymouth Rock	BPR	20	Hiroshima University	15	20
Indigenous chicken overseas	Araucana	ARC	20	Aichi2	20	20
	Black Minorca	BMN	20	Nagoya university, Fukushima7	5	20
	Brahma	BRM	20	Hiroshima University	18	20
	Lite Saussex	LSX	20	Aichi2	14	20
	Belgian Mille Fleur bantam	MIL	2	Hiroshima University	0	2
	White Leghorn-LA	WLL	20	Aichi1	0	20
	White Leghorn-MK	WLM	20	Aichi, Fukushima4, Hiroshima1-2, Mie1, IPLES, AZES, MLRD	0	20
Wild ancestor	Red junglefowl	RJF	20	Nagoya university	0	20
	Total		955		726	831

† FPLES: Fukushima Prefectural Livestock Experimental Station, IPLES: Ibaraki Prefectural Livestock Experiment Station, AZES: Aomori Zootechnical Experiment Station, GPLRI: Gifu Prefectural livestock research institute, APLES: Akita Prefectural Livestock Experiment Station, TUA: Tokyo University of Agriculture, IPAHES: Iwate Prefectural Animal Husbandry Experiment Station, KLES: Kumamoto Livestock Experiment Station, YPLES: Yamaguchi Prefectural Livestock Experiment Station, KPLES: Kochi Prefectural Livestock Experiment Station, NAHES: Niigata Animal Husbandry Experiment Station, MLRD: Mie Livestock Research Division. Prefecture followed by number indicates each chicken fancier.

**Table 2 animals-10-02074-t002:** Haplotypes of Japanese indigenous chicken breeds and chicken breeds from overseas.

Haplotype	Classification *		Total Breeds	Total individuals	Japanese Indigenous Chicken																																						Indigenous Chicken Overseas			
					AJI	CHB	CHN	ENK	GJI	HNI	IJI	JTK	KWC	KNP	KYS	KSM	KMM	KRK	KKS	MJI	MNH	NGY	OHK	OSM	ONG	RJI	SJI	STM	SHK	TKJ	TMR	TSJ	TSK	TTK	UKK	UZR	YKD	YMG	CCN	ING	RIR	BPR	ARC	BMN	BRM	LSX
Hap_1	A02	A	5	22	3					1							13																						2			3				
Hap_2	E01	E	15	91		4				8	4				3					2	1		2						12		1		1	18		9			16		2	8				
Hap_3	C06	C	7	38		26		1								5								1			1									3		1								
Hap_4	A01	A	16	100		17			3			8	13			11		1				14		2	2			6				1		6	2	1	2				11		1		18	1
Hap_5	D22	D	1	3		3																																								
Hap_6	A05	A	12	107		1												12	3		25	3	19	6	25				5	1				4	3								2			
Hap_7	D13	D	8	30		4	3														3		2	5								9			1	3								5		
Hap_8	E12	E	1	1		1																																								
Hap_9	C01	C	11	65		4				2		1				3			3					22		9					9	1			10			1								
Hap_10	C07	C	9	27		2				3								1	3	3		1		6				4	4																	
Hap_11	B01	B	8	51		1							3										3	10				7				5			3					19						
Hap_12	D06	D	5	17		1												6	3											5					2											
Hap_13	D04	D	3	19			16					2								1																										
Hap_14	E09	E	5	11			2			1					1												6							1												
Hap_15	E07	E	3	10					6										2										2																	
Hap_16	E11	E	1	12						12																																				
Hap_17	E03	E	7	22						3		4			5									2										4			1				3					8
Hap_18	C09	C	2	5										4														1																		
Hap_19	E06	E	2	3														2													1												15			
Hap_20	E08	E	1	6											6																															
Hap_21	A03	A	1	1																		1																								
Hap_22	E06_2 ^†^	E	1	1																		1																					2			
Hap_23	A05_2 ^†^	A	1	1																					1																					
Hap_24	E04_2 ^†^	E	1	1																					1																					
Hap_25	D10	D	1	2																				2																						
Hap_26	C11_2 ^†^	C	1	1																				1																						
Hap_27	C11	C	1	2																				2																						
Hap_28	C01_2 ^†^	C	1	2																									2																	
Hap_29	E35	E	1	1																											1															
Hap_30	E06_2 ^†^	E	1	2																											2															
Hap_31	D13_2 ^†^	D	1	3																														3												
Hap_32	D27	D	1	1																															1											
Hap_33	C37	C	1	6																															6											
Hap_34	C02	C	1	1																															1											
Hap_35	E01_2 ^†^	E	1	3																																						3				
Hap_36	E01_3 ^†^	E	1	1																																						1				
Hap_37	E06_3 ^†^	E	1	5																																										5

* Haplotypes are classified according to Miao et al. (2013); ^†^ Newly found haplotypes.

**Table 3 animals-10-02074-t003:** Genetic Diversity indices of Japanese indigenous chickens and indigenous chickens overseas.

D-Loop							Microsatellite DNA Marker							
Category	Breeds	*n*	*h*	Theta-*w*	*Pi*	Tajima’s D	Breeds	*n*	*AR*	*Na*	*Ne*	*Ho*	*He*	*F*
Japanese indigenous chicken	AJI	3	1	0.000	0.000	0.000	AJI	3	1.543	2.185	1.897	0.549	0.427	−0.289
	CHB	64	11	4.441	0.012	1.021	CHB	25	1.623	4.444	2.893	0.433	0.604	0.327
	CHN	21	3	2.224	0.004	−0.149	CHN	24	1.503	3.815	2.183	0.458	0.460	0.004
	ENK	1	1	N.A.	N.A.	N.A.	ENK	1	1.481	1.333	1.333	0.481	0.241	−1.000
	GJI	9	2	2.943	0.008	1.629	GJI	13	1.511	3.111	2.269	0.332	0.468	0.305
	HNI	30	7	4.291	0.008	−0.173	HNI	39	1.509	4.148	2.242	0.379	0.478	0.197
	IJI	4	1	0.000	0.000	0.000	IJI	5	1.506	2.407	1.856	0.363	0.369	0.059
	JTK	15	4	5.228	0.012	0.758	JTK	37	1.577	4.741	2.788	0.496	0.555	0.100
	KWC	16	2	1.507	0.003	0.258	KWC	11	1.525	3.000	2.041	0.392	0.445	0.169
	KNP	4	1	0.000	0.000	N.A.	KNP	4	1.463	1.815	1.522	0.380	0.331	−0.159
	KYS	15	4	0.923	0.002	0.649	KYS	11	1.505	3.148	2.081	0.386	0.441	0.118
	KSM	19	3	3.147	0.011	2.484	KSM	20	1.424	3.185	1.983	0.380	0.414	0.151
	KMM	13	1	0.000	0.000	N.A.	KMM	20	1.517	3.000	2.235	0.565	0.504	−0.118
	KRK	22	5	4.663	0.011	0.848	KRK	19	1.479	3.259	2.043	0.416	0.450	0.101
	KKS	14	5	5.346	0.014	1.192	KKS	25	1.478	3.630	1.979	0.338	0.435	0.279
	MJI	6	3	5.693	0.013	0.830	MJI	10	1.284	2.222	1.529	0.254	0.268	0.007
	MNH	29	3	2.801	0.004	−0.642	MNH	26	1.459	2.889	1.931	0.413	0.431	0.063
	NGY	20	5	5.074	0.005	−1.90 *	NGY	20	1.419	2.778	1.963	0.457	0.408	−0.109
	OHK	26	4	3.669	0.007	−0.122	OHK	31	1.552	3.741	2.504	0.516	0.516	0.013
	OSM	59	11	5.165	0.013	0.867	OSM	53	1.633	6.222	3.054	0.537	0.593	0.091
	ONG	29	4	2.037	0.001	−2.048 *	ONG	11	1.512	2.630	1.799	0.421	0.406	−0.030
	RJI	9	1	0.000	0.000	N.A.	RJI	9	1.263	1.593	1.386	0.184	0.195	0.087
	SJI	7	2	3.673	0.005	−1.594	SJI	10	1.317	2.481	1.793	0.278	0.289	0.154
	SHK	25	5	4.502	0.012	1.272	SHK	23	1.440	3.037	2.094	0.399	0.428	0.060
	STM	18	4	4.070	0.012	2.001 *	STM	21	1.580	4.074	2.540	0.457	0.546	0.195
	TKJ	6	2	4.380	0.007	−1.435	TKJ	6	1.473	2.444	1.840	0.373	0.389	0.086
	TMR	14	5	3.459	0.009	1.443	TMR	8	1.513	2.889	2.048	0.352	0.428	0.200
	TSJ	16	4	4.520	0.010	0.567	TSJ	35	1.479	3.963	2.353	0.358	0.464	0.233
	TSK	1	1	N.A.	N.A.	N.A.	TSK	1	1.333	1.074	1.074	0.333	0.167	−1.000
	TTK	36	6	3.376	0.009	0.937	TTK	51	1.511	4.074	2.287	0.439	0.476	0.085
	UKK	29	9	4.583	0.012	1.053	UKK	25	1.646	5.296	3.081	0.552	0.619	0.113
	UZR	16	4	4.219	0.008	−0.018	UZR	11	1.536	3.185	2.216	0.462	0.470	0.034
	YKD	3	2	6.000	0.012	N.A.	YKD	1	1.296	1.037	1.037	0.296	0.148	−1.000
	YMG	2	2	2.000	0.004	N.A.	YMG	0	−	−	−	−	−	−
	CCN	18	2	2.617	0.004	−0.994	CCN	20	1.422	3.296	1.977	0.463	0.411	−0.097
	ING	19	1	0.000	0.000	N.A.	ING	20	1.351	2.296	1.730	0.291	0.342	0.142
	RIR	16	3	2.712	0.008	1.732	RIR	20	1.481	2.926	2.192	0.456	0.469	0.045
	BPR	15	4	3.075	0.007	0.106	BPR	20	1.464	3.074	2.312	0.463	0.452	0.004
Indigenous chicken overseas	ARC	20	4	3.382	0.002	−0.423	ARC	20	1.326	2.000	1.691	0.348	0.318	−0.100
	BMN	5	1	0.000	0.000	0.000	BMN	20	1.297	2.148	1.585	0.148	0.289	0.543
	BRM	18	2	0.291	0.000	−1.165	BRM	20	1.260	2.407	1.736	0.391	0.376	−0.039
	LSX	14	3	3.145	0.005	−0.737	LSX	20	1.385	1.667	1.465	0.324	0.254	−0.256
							MIL	2	1.352	1.630	1.519	0.370	0.241	−0.538
							WLL	20	1.417	2.778	1.931	0.402	0.406	0.047
							WLM	20	1.410	2.593	1.870	0.396	0.400	0.013
Wild ancestor							RJF	20	1.484	2.704	2.017	0.513	0.472	−0.075

* *p* < 0.05.

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
