# Peer review of "Geographic Origin and Genetic Characteristics of Japanese Indigenous Chickens Inferred from Mitochondrial D-Loop Region and Microsatellite DNA Markers"

_animals, 2020, doi:10.3390/ani10112074_

Round 1

Reviewer 1 Report

I am satisfied with the responses to my concerns. The manuscript can be accepted in present form.

Author Response

Open Review

General remarks

In this type of phylogenetic and genetic characterization, the results concerning only one animal /ENK, TSK/ do not make sense.

Lines453-454, 505-507: We have noted that we could not examine sufficient number of individuals for several breeds to understand genetic diversities and phylogenetic relationships. However, genetic characterizations for ENK and TSK have never been performed so far. So we would like to include genotype data of these breeds as a preliminary data for further detailed study in the future.

Specific remarks

In Methods, some of the data needs to be clarified

Clear information on the number of investigated chicken breeds is necessary. Line 154 and 158:..seven chicken breeds from abroad. However in the table 1. there are 8 indigenous chicken oversea.

One of the 8 chicken populations in Tables was a population of wild ancestor, the Red junglefowl. So we did not count the population as indigenous chicken breeds. We have modified the main text (Line 160) and Table 1 to be more understandable.

It would be good to standardize the nomenclature: "chicken breeds from abroad" or "indigenous chicken oversea".
We have standardized the nomenclature as “chicken(s) oversea” for exotic breeds used in this study, such as ARC, BRM, and LSX, in the revised manuscript.

Information on the number of  characterized birds is not consistent.
We have carefully checked numbers in the manuscript, such as the number of individuals and genetic diversity indices.

Line 391 ".... 812 individuals...., and the different numbers were cited  in the Table 1.
We have checked the numbers of individuals examined by D-loop and microsatellite markers and corrected them.

Reviewer 2 Report

In the revised version of the manuscript the authors took into account all my previous comments.

I think this manuscript will be acceptable now.

Author Response

Open Review

This study analyzed geographic origin and genetic characteristics of Japanese indigenous chickens based on mitochondrial D-loop region and microsatellite markers. It had some reference values. While according to the number of individuals in each breed, the investigated individuals in many chicken breeds were very limited (Table 1). For example, in the experiment of mitochondrial D-loop region, the number of individuals in 8 chicken breed just ranged from 1 to 5. In the experiment of microsatellite markers, the investigated individuals in 7 chicken breeds less than 6. The limited number of individuals in many chicken breeds will affect the geographic origin and genetic characteristics of Japanese indigenous chickens.

As the Reviewer pointed, we could not examine sufficient numbers of individuals or populations for several breeds to fully understand the histories and genetic diversities of Japanese indigenous chickens. We have added the notion in Discussion and Conclusion (Lines 451-452, Lines 503-505).

Besides, there are several minor mistakes in the present manuscript.

  1. Line 44, Line 181. The authors stated that 27 microsatellite markers were used in this study, while 28 microsatellite markers were listed in Table S1.
    The number of microsatellite markers was 27. We have deleted one marker “MCW0014” from Table S1, which was not used for this study.

  2. Line 287, according to present manuscript, 9 chicken breed had one haplotype.
    Actually nine chicken breeds had one haplotype but one of the breeds,,BRM (Brahma), is not Japanese indigenous chicken. We have corrected the sentence to be more understandable by adding a phrase “Among the Japanese indigenous chicken breeds …” at Line 292.

  3. Line 303, “Japan (n = 944)”, while in figure 2, the listed number was 726/938. Why it showed two numbers in the figure 2.
    Line 310: We mistakenly included chickens breeds overseas in the number. We have corrected the number of individuals.

  1. Line 327-328. what is the meaning of this sentence “and 11 of the 27 markers showed more than three populations.”?
    Lines 333-336: The sentence was described insufficiently and redundantly. We have modified the sentences.

  2. Line 331-332. The authors stated that YKD exhibited the least genetic diversity. According to Table 2, only 1 individual was analyzed for this chicken breed. This may be the reason why YKD exhibited the least genetic diversity. So the limited number will affect the results which I stated at the beginning.
    Lines 339-349: We have compared genetic diversities of 34 Japanese indigenous chicken breeds without ENK, TSK, YKD and YMG.

  3. Line 363-364. “the STRUCTURE analysis of the 38 Japanese indigenous chicken breeds indicated K=2 as the highest (delta K= 2.05). ”According figure 4c, delta K= 2.05 should be wrong.
    The delta K at K=2 was “5.42”. We have corrected it (Line 375).

  4. Line 387-389. The authors stated that “To date, the mitochondrial D-loops of 23 breeds have been examined [9,11,23,32,33,66,67], and 24 breeds have been examined using microsatellite DNA markers [34-36]”. References 34-36 were published in 2006-2008, more than 10 years passed, how can the authors come the conclusion. I also have the same doubt about the expression “To date, the mitochondrial D-loops of 23 breeds have been examined”.
    Lines 398-403: We have clarified the sentences was a description for “Japanese indigenous chicken breeds”. The numbers of Japanese indigenous chicken breeds that have been examined for the mitochondrial D-loop and microsatellite markers so far are 23 and 24, respectively. Genetic diversities among the Japanese indigenous chicken breeds were well studied in many researches that were cited in the manuscript, while genetic relationships of the indigenous chicken breeds to chicken breeds overseas remained unveiled to date.

This manuscript is a resubmission of an earlier submission. The following is a list of the peer review reports and author responses from that submission.

Round 1

Reviewer 1 Report

The aim of the study was the genetic characterization of Japanese indigenous chicken breeds and chicken breeds from abroad using mitochondrial D-loop region and 27 microsatellite DNA markers.

It is an interesting analysis performed with the use of appropriate tools

General remarks

In this type of phylogenetic and genetic characterization, the results concerning only one animal /ENK, TSK/ do not make sense.

Specific remarks

In Methods, some of the data needs to be clarified

Clear information on the number of investigated chicken breeds is necessary. Line 154 and 158:..seven chicken breeds from abroad. However in the table 1. there are 8 indigenous chicken oversea.

It would be good to standardize the nomenclature: "chicken breeds from abroad" or "indigenous chicken oversea".

Information on the number of  characterized birds is not consistent.

Line 391 ".... 812 individuals...., and the different numbers were cited  in the Table 1.

Reviewer 2 Report

This study analyzed geographic origin and genetic characteristics of Japanese indigenous chickens based on mitochondrial D-loop region and microsatellite markers. It had some reference values. While according to the number of individuals in each breed, the investigated individuals in many chicken breeds were very limited (Table 1). For example, in the experiment of mitochondrial D-loop region, the number of individuals in 8 chicken breed just ranged from 1 to 5. In the experiment of microsatellite markers, the investigated individuals in 7 chicken breeds less than 6. The limited number of individuals in many chicken breeds will affect the geographic origin and genetic characteristics of Japanese indigenous chickens. Besides, there are several minor mistakes in the present manuscript.

  1. Line 44, Line 181. The authors stated that 27 microsatellite markers were used in this study, while 28 microsatellite markers were listed in Table S1.
  2. Line 287, according to present manuscript, 9 chicken breed had one haplotype.
  3. Line 303, “Japan (n = 944)”, while in figure 2, the listed number was 726/938. Why it showed two numbers in the figure 2.
  4. Line 327-328. what is the meaning of this sentence “and 11 of the 27 markers showed more than three populations.”?
  5. Line 331-332. The authors stated that YKD exhibited the least genetic diversity. According to Table 2, only 1 individual was analyzed for this chicken breed. This may be the reason why YKD exhibited the least genetic diversity. So the limited number will affect the results which I stated at the beginning.
  6. Line 363-364. “the STRUCTURE analysis of the 38 Japanese indigenous chicken breeds indicated K=2 as the highest (delta K= 2.05). ”According figure 4c, delta K= 2.05 should be wrong.
  7. Line 387-389. The authors stated that “To date, the mitochondrial D-loops of 23 breeds have been examined [9,11,23,32,33,66,67], and 24 breeds have been examined using microsatellite DNA markers [34-36]”. References 34-36 were published in 2006-2008, more than 10 years passed, how can the authors come the the conclusion. I also have the same doubt about the expression “To date, the mitochondrial D-loops of 23 breeds have been examined”.
